IPPP/24/30

# Photoproduction in general-purpose event generators

I. Helenius[†1,2], P. Meinzinger[‡3*], S. Plätzer[§4,5], and P. Richardson[¶3]

[1]University of Jyvaskyla, Department of Physics, P.O. Box 35, FI-40014 University of Jyvaskyla, Finland
[2]Helsinki Institute of Physics, P.O. Box 64, FI-00014 University of Helsinki, Finland
[3]Institute for Particle Physics Phenomenology, Durham University, Durham DH1 3LE, UK
[4]Institut für Physik, NAWI, Universität Graz, Universitätsplatz 5, 8010 Graz, AT
[5]Teilchenphysik, Fakultät für Physik, Universität Wien, Boltzmanngasse 5, 1030 Wien, AT

**Abstract:** We present a comparison of three different general-purpose Monte Carlo event generators, HERWIG, PYTHIA, and SHERPA, with respect to the simulation of photoproduction. We outline the default inputs, implementation differences and compare the results at different stages of the event generation. We find that, despite a similar starting point, the final cross sections do have some differences related to different non-perturbative inputs. We compare the simulations with experimental data for jet production in LEP and HERA and find that all generators provide a decent desription of the data within the considered uncertainties. We also present predictions for the upcoming EIC for jet observables and event shapes and conclude that accurate simulations will require further phenomenological advances.

## 1 Introduction

General-purpose Monte Carlo (MC) event generators [1] provide computational tools for complete simulations of high-energy collisions with different beam configurations. They model broad ranges of physics and energy scales starting from a hard partonic scattering down to hadronization and decays of unstable hadrons. Thus these generators can be applied to quantify what kind of particle distributions a specific partonic scattering would result in a particle detector and use to connect a theoretical first-principle calculation to an experimental measurement. Accurate and realistic simulations are crucial for planning and performing experimental analyses with a prime example being the detector collaborations at the LHC applying these programs virtually in all analyses.

The main players in the field of general-purpose event generators are HERWIG [2,3], PYTHIA [4] and SHERPA [5,6], each including a long history of development over several decades. The main theoretical framework that the generators built upon consists of the collinear factorization and perturbative Quantum Chromodynamics

---

[†]ilkka.m.helenius@jyu.fi
[‡]peter.meinzinger@durham.ac.uk
[§]simon.plaetzer@uni-graz.at
[¶]peter.richardson@durham.ac.uk

(pQCD) [7] that can be used to calculate differential cross section of varying perturbative order for relevant partonic scatterings. The radiation of the initial and final partons participating to the hard scattering can be modelled using DGLAP evolution equations [8–11] and parton-showers algorithms [12] that turn the inclusive evolution equations into exclusive emission probabilities using Sudakov factors. In case of beam particles with partonic structure a complete partonic final state require also modelling multiparton interactions (MPIs) [13], that generates the underlying event present in addition to hard scattering of interest, and beam remnants to account for conservation of energy and quantum numbers. Finally the partons carrying colour charge are turned into colour-neutral particles applying non-pertubative hadronization [14, 15] and colour reconnection models [16, 17].

The development of the current event generators have mainly been driven by the studies of proton-proton (pp) collisions at the LHC. The provided luminosities have enabled measurements to reach accuracies of a few percent-level, necessitating persisting theoretical computations of the differential cross-sections to matching or higher precision. As an example of the latter, matching parton shower consistently with matrix-element computations at next-to-leading order (NLO) [18, 19] have been developed and implemented in all major event generators and for some particluar cases even beyond NLO [20–22]. Another avenue where recent progress has been made is to extend the modern generators to handle other beam configurations including heavy-ion collisions [23] and electron-proton (ep) collisions [21, 24–27]. For the latter the main motivation is provided by the electron-ion collider (EIC) that is the next large high-energy collider project pushed forward at BNL in US [28]. The ep collisions can be classified in terms of $Q^2$, the virtuality of intermediate photon, which is related to the kinematics of the scattered electron. In deep inelastic scattering processes (DIS) the exchanged photon is considered as a virtual particle scattering with a single quark inside the proton. In the photoproduction limit, $Q^2 \to 0$, the photon may, however, fluctuate into a hadronic state where the underlying hard scattering takes place between two coloured partons and are thus similar to a collision between two hadrons. In fact this resolved component dominates the total cross section of photon-proton ($\gamma$p) system and has to be accounted for in addition to contributions of a direct photon scattering. Similarly one can study photon-photon ($\gamma\gamma$) collisions in electron-positron (e$^+$e$^-$) annihilation.

In this study we present comparisons of the three general purpose event generators, PYTHIA, HERWIG and SHERPA, with respect to photoproduction. In Sec. 2 we present the frameworks for each of the generators and discuss their differences. In Sec. 3 and 4 we present cross sections for dijet photoproduction and compare the results with the LEP and HERA data. We also show predictions for dijet photoproduction at kinematics relevant to the EIC including comparisons to event-shape and inclusive QCD observables in Sec. 5. We conclude our findings and provide a brief outlook in Sec. 6.

## 2   Event Generators

For photons resolved into a hadronic state, the partonic structure can be described with DGLAP-evolved parton distribution functions (PDFs), $f_i^\gamma(x_\gamma, \mu_{\rm F}^2)$, where $x_\gamma$ is the momentum fraction of a given parton wrt. the photon and $\mu_{\rm F}^2$ is the factorization scale at which the PDFs are evaluated at. Similarly as the PDFs for protons, $f_i^{\rm p}(x_{\rm p}, \mu_{\rm F}^2)$, the photon PDFs are obtained in a global analysis where the parameters of a non-perturbative input are fitted into experimental data. In addition to a fitted hadron-like part, the photon PDFs contain also a point-like (anomalous) contribution from perturbative $\gamma \to q\bar{q}$ splittings that has to be included in the evolution equations when comparing to experimental data. The structure of the resolved photons were an active topic around 20 years ago when LEP was producing experimental data for such analyses but, since then, activities have been sparse and the available fits are lacking the recent methodological developments, e.g. for the uncertainty estimation.

Knowing the structure of the incoming states, one can factorize hard-process generation for photoproduction in ep collisions as

$$d\sigma_{\rm hard}^{\rm ep} = f_\gamma^{\rm e}(x, Q^2) \otimes f_i^\gamma(x_\gamma, \mu_{\rm F}^2) \otimes f_j^{\rm p}(x_{\rm p}, \mu_{\rm F}^2) \otimes d\sigma_{\rm hard}^{ij}, \tag{2.1}$$

where $f_\gamma^{\rm e}(x, Q^2)$ is the photon flux from an electron at a given momentum fraction $x$ and photon virtuality $Q^2$, and $\sigma_{\rm hard}^{ij}$ perturbatively calculable hard coefficient function for a given hard process with initiating particles $i$ and $j$. In case of direct-photon scattering the photon will act as an initiating particle and there one can replace the $f_i^\gamma$ with a delta function. Thus, there are two contributions that should be accounted: direct and resolved. At leading order (LO) these contributions can be taken as separate but at higher orders one has to make sure that no double counting takes place when combining these two event classes.

The photon flux can be computed using equivalent photon approximation (EPA) [29]. For photons from a charged lepton $l$ it gives in the leading-log (LL) approximation

$$f_\gamma^l(x) = \frac{\alpha_{\text{em}}}{2\pi} \frac{1 + (1-x)^2}{x} \log \left[ \frac{Q_{\text{max}}^2}{Q_{\text{min}}^2(x, m_l)} \right]. \tag{2.2}$$

The lower limit for the virtuality follows from kinematics and the upper limit is adjusted according to the experimental configuration.

In case of $\gamma\gamma$ collisions in $e^+e^-$ the corresponding factorization reads

$$d\sigma_{\text{hard}}^{\text{ee}} = f_\gamma^{\text{e}}(x_1, Q_1^2) \otimes f_i^\gamma(x_\gamma^+, \mu_{\text{F}}^2) \otimes f_\gamma^{\text{e}}(x_2, Q_2^2) \otimes f_j^\gamma(x_\gamma^-, \mu_{\text{F}}^2) \otimes d\sigma_{\text{hard}}^{ij}. \tag{2.3}$$

In this case there are four different contributions, direct-direct, direct-resolved, resolved-direct and resolved-resolved that need to be accounted for. This factorized approach provides the starting point for all considered event generators over which further modelling of parton showers, beam remnants, MPIs and hadronization are included. Thus, even though the starting point is the same for all generators, the results will depend on the applied inputs and particular choices made in this modelling. Before comparing the different event generators to data, we will therefore describe the main features of the generators and the default inputs, and highlight some significant differences in the following subsections.

## 2.1 HERWIG

HERWIG [2,3,30] is a multi-purpose event generator, which has traditionally been focusing on accurate QCD simulations. It includes two parton shower modules, an angular ordered one based on coherent branching [31], and more recently a dipole parton shower [32,33]. Both parton shower modules include full mass effects, spin correlations, and initial and final state radiation [34,35]. Hard processes can be simulated using a wide range of built-in matrix elements at leading and next to leading order, and via the Matchbox module [3,33] using external matrix element providers and an automated matching to NLO QCD either within the MC@NLO or POWHEG approaches. Multi-jet merging [36,37] is available using the dipole shower. Photoproduction processes can be simulated in $e^+e^-$, $ep$ and $pp$ collisions including all direct and resolved components. The PDF sets and fluxes we currently provide are the SaS photon PDFs [38] with the 1D, 1M, 2D and 2M schemes, and photon fluxes following a Weizsäcker-Williams parametrization for electrons, and the Budnev approach [39] for protons. Hadronization is modelled using the cluster hadronization model [2], and we provide an eikonal MPI model, which currently however is limited to $pp$ collisions and unfortunately cannot yet treat resolved photons due to technical issues. Similar restrictions currently apply to NLO matching using Matchbox, though both of these issues are currently being worked on and should be available with one of the next HERWIG releases. In order to assess uncertainties, we can vary the scales of the hard process as well as the hard veto scales of the parton showers, which we perform in this study, additionally to varying the photon PDF sets. All of these variations turn out not to be significant, with the PDF set variation contributing the biggest uncertainty with similar findings as reported in [2]. The HERWIG parton showers always need to terminate on a valence quark of the incoming parton, and thus create a remnant which complements the parton which had been extracted. In case of the photon we will therefore always generate an anti-quark remnant, which might cause additional (collinear) shower activity even if radiation in the showers is switched off entirely.

## 2.2 PYTHIA

Simulations for photoproduction in ep and $\gamma\gamma$ collisions in $e^+e^-$ have been fully enabled from release 8.226 onwards and the current setup is described in the 8.3 manual (Ref. [4]). Both hard and soft QCD processes can be generated, the former including processes like inclusive and diffractive jet production, and the latter non-diffractive and diffractive events and also elastic scattering. The implementation of resolved processes is based on CJKL photon PDFs [40] which includes hadron-like part with non-perturbative input separately from the perturbatively calculated anomalous contribution. In the generation of resolved processes these are not considered separately but the parton-shower algorithm includes the $\gamma \to q\bar{q}$ splitting that may collapse the resolved photon into an unresolved state at a perturbative scale corresponding to the anomalous part of the evolution equation.

The simulations include generation of MPIs as long as the photons are in a resolved state and the MPI framework, where the QCD cross sections are regulated with the screening parameter $p_{T,0}$, can be applied also for non-diffractive soft QCD processes without any phase-space cuts. The cross section for the different soft processes are obtained from the SaS parametrizations in Ref. [38]. For proton PDFs the current default NNPDF23_lo_as_0130_qed has been applied and the value of $\alpha_S$ has been fixed accordingly to 0.130 at $\mu_R^2 = M_Z^2$. Here we have used PYTHIA version 8.310 released in July 2023.

In this study we have applied the default setup for photoproduction in PYTHIA 8.3. In case of $\gamma\gamma$ a specific tune for $p_{T,0}$ has been applied based on single-inclusive charged-particle production data from OPAL at LEP [41] and in case $\gamma$p the $p_{T,0}^{\mathrm{ref}}$ in the standard parametrization has been adjusted to 3.0 GeV based on comparisons to single-inclusive charged particle production in photoproduction at HERA [42] which is in line also with the multiplicity distributions measured recently by ZEUS in Ref. [43]. Heavy quark masses are included apart from the $gQ \rightarrow gQ$ subprocess where massless expressions for the matrix element are applied also for charm and bottom quarks. The remnants are constructed by adding a minimal number of partons such that momentum and colour are conserved. In case of resolved photons the flavour for "valence" quark-antiquark pair is sampled according to relative fractions of the PDFs at the specific kinematics and part of the phase-space is cut out to allow room for the remnant partons. The scale variations have been performed by varying renormalization and factorization scales independently in the hard process generation by a factor of two, excluding the variations where the relative difference would be 4.

## 2.3 SHERPA

In the SHERPA event generator [5,6], photoproduction of jets at MC@NLO accuracy has been recently implemented and validated against data [26] and will be published under version 3.0. Resolved processes can be calculated through interfaces to various different photon PDFs, where the default is the SaS PDF library [38].

We generated events at Leading Order and MC@NLO accuracy as described in [26]. We used AMEGIC [44] and COMIX [45] for tree-level matrix elements and subtraction terms [46], and OPENLOOPS [47] for one-loop matrix elements. For the calculation of the matrix element, we treated the $b$-quarks for ZEUS, and additionally the $c$-quark for OPAL and EIC runs, as massive and included them in the final state. While at LO all subprocesses are available, the initial state subtraction for massive quarks has not been implemented yet. The CSSHOWER parton shower [48] was used, combined with the MC@NLO method [18,49] as implemented in SHERPA [50].

Multiple-parton interactions were modelled through an implementation of the Sjostrand-van Zijl model [51, 52] in which the total hadronic cross-section is calculated using Regge theory and the parametrizations for the photon are obtained as a superposition of light neutral vector mesons as proposed and parametrised in [52]. The tuning of the rewritten MPI modelling is currently work-in-progress as part of the new release. Particles were hadronized through the cluster fragmentation model in AHADIC [53] and has already been tuned against data. The photon flux was modelled as computed in [54], which includes a correction for $x \rightarrow 1$ relevant for lepton-hadron colliders. For the PDFs, we used the SAS2M [55,56] set for the photon and the PDF4LHC21_40_pdfas [57] set for the proton, and the value of $\alpha_S$ was kept at its default value of 0.118 with three-loop running. Factorisation and renormalisation scale were set to $\mu_F = \mu_R = H_T/2$ and a 7-point scale variation in the matrix element and the shower was done as an uncertainty estimation.

## 2.4 Differences between generators

As a baseline for a broader comparison between the generators, we compared events of LEP-like setups with beam energies of 99 GeV from PYTHIA and SHERPA at the bare cross-section-level for each of the components. Here we used the same photon flux, PDF, and a setting of $\alpha_S = 0.130$ with 1-loop running and computed only processes of light partons. In Fig. 1 we can see that indeed the results are, up to statistical fluctuations, in a perfect agreement. The considered observables include the energy and pseudorapidity distribution, and transverse momentum spectra of the outgoing partons for direct, single-resolved and double resolved contributions separately. In case of single-resolved contribution the direct photon has a positive $p_z$.

In Tab. 1, we summarise the differences between the generators with respect to the simulation of photoproduction. Other differences are for example the fragmentation model and QED corrections, however, these will not be discussed here and can be found in the manuals [2, 4, 58].

Based on this, we changed the beam configuration on the previous setup to an EIC-like setup with electron-proton beams with 18 and 275 GeV, and looked at photoproduction events with resolved photons. We used

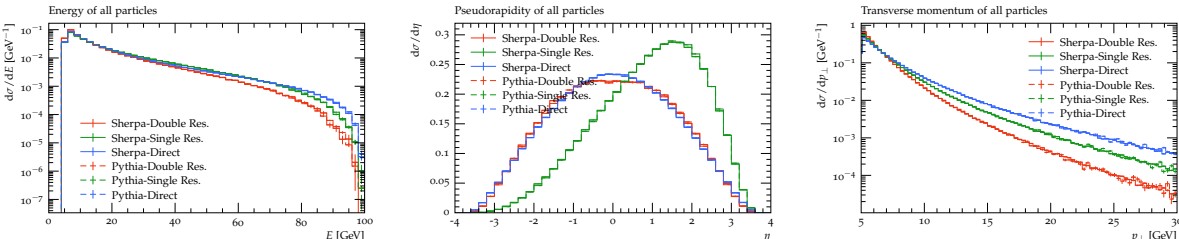

**Figure 1:** Comparison of fixed-order parton level events between PYTHIA and SHERPA for a LEP-like setup.

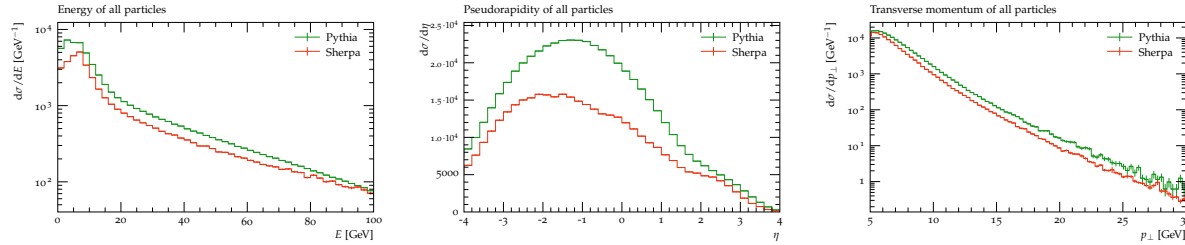

**Figure 2:** Comparison of fixed-order parton level events including ISR parton shower and beam remnants between PYTHIA and SHERPA for a EIC-like setup.

the same settings as before, however, now also modelling the Initial State Radiation (ISR) and the beam remnants, including charm and bottom quarks in the processes and using the NNPDF23_lo_as_0130_qed PDF for the proton. The total cross-section in PYTHIA is larger than in SHERPA due to the way the remnants are created: in PYTHIA, forced splittings are combined with phase space rejection to ensure the correct creation of the valence quark content of the beam particle, which allows for an iterative procedure. In SHERPA, the remnant creation is purely based on phase space rejection which naturally will lead to more events being rejected, especially for charm and bottom quarks. In the subsequent modelling of the beam breakup, we can see that SHERPA creates more activity in the low-$E_T$ and low-$p_T$ region, whereas PYTHIA tends to have higher transverse momenta and energies in its particle spectrum.

Coming back to the modelling of the perturbative part, the event generators model the flux to different accuracies where the difference is given by

$$f_{\text{NLL}}(x) - f_{\text{LL}}(x) = -\frac{\alpha_{\text{em}}}{\pi} m_e^2 x \left( \frac{1}{Q_{\min}^2} - \frac{1}{Q_{\max}^2} \right) \ . \tag{2.4}$$

The correction comes with an overall negative sign and hence will lead to smaller total cross-sections. In differential observables, the correction will however only be sizeable in regions where $x \to 1$. Even though not specific to photoproduction we point out that the settings of $\alpha_S$ are different: while in SHERPA the current default is set according to the default proton PDF set and close to the PDG world average [59], in PYTHIA it is being used as a dynamical $K$ factor, also in accordance with its default proton PDF set. Photon PDFs do not allow the nowadays standard procedure of setting $\alpha_S$ in accordance with the fit as this information is often not given in the corresponding publication and would in any case be in conflict with settings of modern proton PDFs. Hence, strictly speaking the factorisation of the cross section is not fully consistent due to the different values of $\alpha_S$ used throughout event generation.

As a last point, we plot the parton distributions of the light partons for the two PDF libraries SAS2M and CJKL in Fig. 3. The two fits come to vastly different behaviour at small $x$, where the SAS2M sets reach an almost constant value and the CJKL sets behave like a power-law, with the most pronounced difference in the gluon distribution.

It is also worthy to point out that unlike the settings for $\alpha_S$ and the flux, which cover the perturbative region and are independent of any fitting, the remnant creation and the MPIs interplay with non-perturbative effects and rely on precise data by experiments like HERA and LEP.

| Property | PYTHIA | SHERPA | HERWIG |
|---|---|---|---|
| Flux | LL | NLL | LL |
| $\alpha_S(M_Z^2)$ | 0.130, 1-loop running | 0.118, 3-loop running | 0.118, 2-loop running (hard process) |
| PDFs | CJKL | SAS2M | SAS2M |
| Remnants | forced splittings/PS rejection | PS rejection | forced splitting |
| Photon Splitting | yes | no | no |
| MPI tuning | preliminary $\gamma$p/$\gamma\gamma$ tune | untuned | untuned |

**Table 1:** Differences between the generators of default settings, specifically for photoproduction.

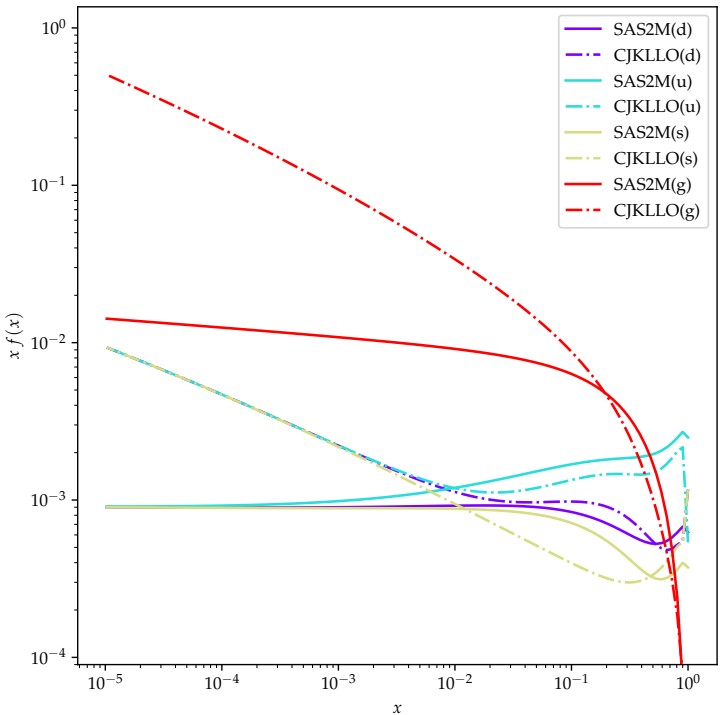

**Figure 3:** Comparison of the CJKLLO and SAS2M PDF sets for light quarks and the gluon at $\mu_\mathrm{F}^2 = 5 \ \mathrm{GeV}^2$.

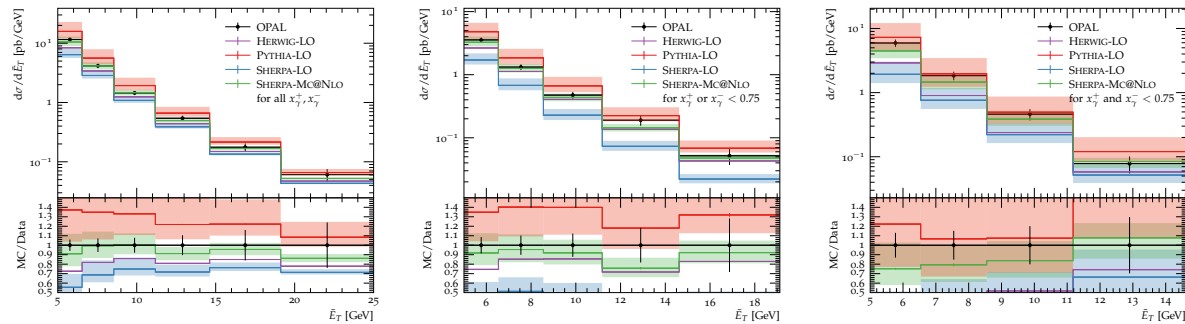

**Figure 4:** Distribution of average transverse energy of di-jets, $\bar{E}_T$, from OPAL [60] for all $x_\gamma^\pm$ (left), $x_\gamma^+$ or $x_\gamma^- < 0.75$ (middle) and $x_\gamma^\pm < 0.75$ (right), compared to Leading Order simulations by HERWIG, PYTHIA and SHERPA and MC@NLO-accurate simulations by SHERPA.

## 3 Comparisons to LEP

For the comparison to LEP data, we used a dijet measurement from the OPAL collaboration [60]. At lepton colliders, the cross-section for photoproduction can be separated into $\sigma = \sigma_{\gamma\gamma \to X} + \sigma_{\gamma j \to X} + \sigma_{j\gamma \to X} + \sigma_{jj \to X}$, where $j$ denotes a photon being resolved into partons. The analysis used data taken at $\sqrt{s} = 198$ GeV and clustered jets with the $k_T$ algorithm with $R = 1$ demanding $E_T > 3$ GeV and $|\eta| < 2$ with at least two jets. To separate resolved from direct photoproduction processes, the analysis defined

$$x_\gamma^\pm = \frac{\sum_{j=1,2} E^{(j)} \pm p_z^{(j)}}{\sum_{i \in \text{hfs}} E^{(i)} \pm p_z^{(i)}} \tag{3.1}$$

and associated values $x_\gamma^\pm < 0.75$ with resolved processes. For a parton-level $2 \to 2$ scattering these definitions would match with the momentum fractions in photons going positive and negative $p_z$ in Eq. 2.3 but adding parton showers, MPIs and hadronization will smear the kinematics such that the division will be only approximative. In the numerator the sum runs over the two jets with the highest transverse momentum, $p_T$, and in the denominator over the complete hadronic final state. In Fig. 4 we present the average transverse energy of the di-jets for different taggings on $x_\gamma$. The LO prediction from SHERPA undershoots the total cross-section, however, the effect is more pronounced for resolved photons than for unresolved ones. Going to NLO accuracy, the simulation describes the data well within the errors. The large $K$ factors hint at the real corrections and the filled-up phase space as the drivers of the improved description. PYTHIA tends to overshoot the data for events with direct contribution but agree well with the data for the resolved-resolved case. The HERWIG simulation typically lies between the SHERPA and PYTHIA results. In all cases the data is within the large uncertainties from scale variations. As discussed in Sect. 2.4, the large difference between the LO results from HERWIG, PYTHIA and SHERPA builds up from varying inputs.

The distributions in pseudo-rapidity $\eta$ in Fig. 5 show a similar picture, however for double-resolved processes, *i.e.* $x_\gamma^\pm < 0.75$, all predictions have a slightly steeper fall-off as a function of $\eta$ than seen in the data, leaning towards an undershoot in the forward region. One potential reason could be the weakly constrained gluon-content of the photon, which should be the leading contribution at low-$E_T$ and forward jets.

Fig. 6 shows distributions of $x_\gamma$ for low and high average jet transverse energies $\bar{E}_T$, respectively. We see good agreement for both the PYTHIA and the SHERPA-MC@NLO simulations; however, the transition from the resolved to the direct processes seems to be poorly modelled, as can be seen by the consistent undershoot at around $x_\gamma \approx 0.8$ for all predictions. Unlike HERWIG and SHERPA, the PYTHIA simulation does include the correct evolution of the photon PDF, *i.e.* the photon splitting $\gamma \to q\bar{q}$ is taken into account, the distribution still shows this shape in that case too, though. Potential reasons could be the poor quality of the photon PDF or insufficient tuning of the fragmentation and multiple-parton interactions modelling. Combined with the overshoot in the largest-$x$ bin, we would expect these effects to increase the multiplicity of direct processes, hence shifting cross-section towards lower values of $x$.

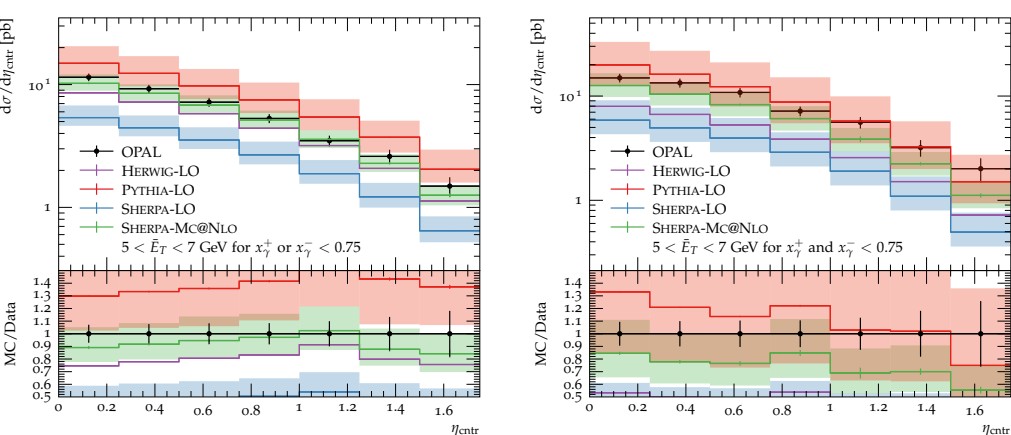

**Figure 5:** Distribution of pseudo-rapidity $\eta$ of di-jets from OPAL [60] for $x_\gamma^+$ or $x_\gamma^- < 0.75$ (left) and $x_\gamma^\pm < 0.75$ (right), compared to Leading Order simulations by HERWIG, PYTHIA and SHERPA and MC@NLO-accurate simulations by SHERPA.

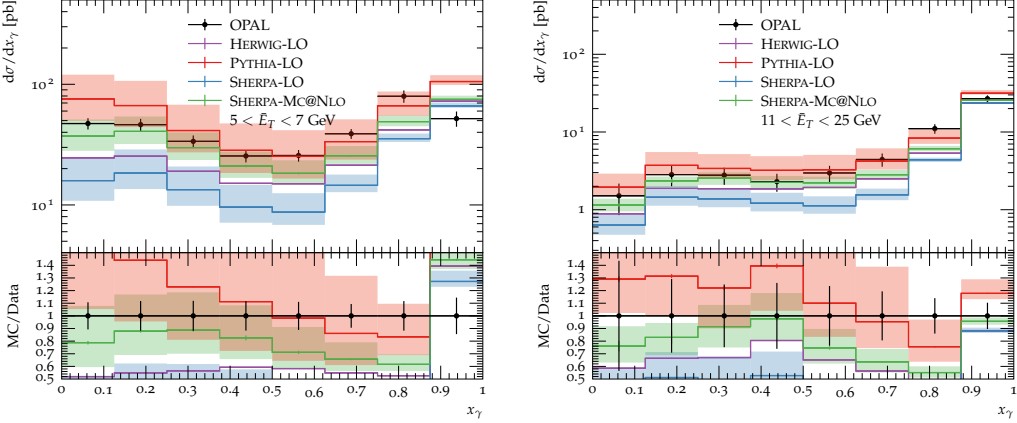

**Figure 6:** Distribution of $x_\gamma$ of di-jets from OPAL [60] for bins of average jet transverse energy $\bar{E}_T \in [5\,\mathrm{GeV}, 7\,\mathrm{GeV}]$ (left) and $\bar{E}_T \in [11\,\mathrm{GeV}, 25\,\mathrm{GeV}]$ (right), compared to Leading Order simulations by HERWIG, PYTHIA and SHERPA and MC@NLO-accurate simulations by SHERPA.

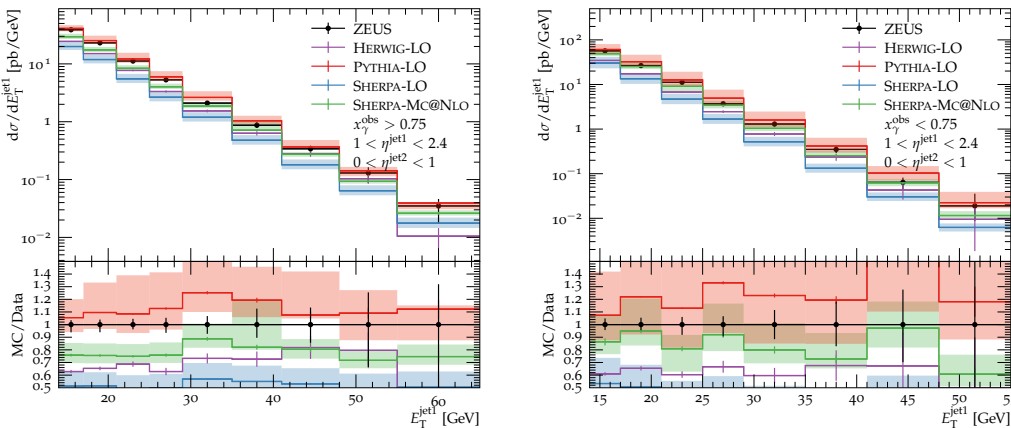

**Figure 7:** Distributions of leading jet transverse energy $E_T^{\text{jet1}}$ for $x_\gamma > 0.75$ (left) and $x_\gamma < 0.75$ (right) with jet pseudo-rapidities for the two leading jets in $1 < \eta^{\text{jet1}} < 2.4$ and $0 < \eta^{\text{jet2}} < 1$ from ZEUS [61], compared to Leading Order simulations by HERWIG, PYTHIA and SHERPA and MC@NLO-accurate simulations by SHERPA.

## 4 Comparisons to HERA

For this comparison we used data taken by the ZEUS collaboration [61] during HERA Run 1. Here, the photoproduction cross-section can be decomposed into two parts, $\sigma = \sigma_{\gamma P \to X} + \sigma_{jP \to X}$, where again $j$ denotes a parton resolved from within the photon. Similar to the previous analysis, jets were clustered with the $k_T$ algorithm with $R = 1$ with cuts of $\eta \in [-1, 2.4]$ and $E_T > 14$ (11) GeV for the (sub-)leading jet, respectively. The analysis used a similar discriminant for the tagging of direct and resolved processes, defined as

$$x_\gamma^{\text{obs}} = \frac{\sum_{j=1,2} E_T^{(j)} \mathrm{e}^{-\eta^{(j)}}}{2yE_e} \, , \tag{4.1}$$

where the sum runs over two highest $E_T$ jets, $E_e$ is the energy of the incoming electron and $y$ the inelasticity. In Fig. 7 we show the leading jet transverse energy $E_T^{\text{jet1}}$ for direct and resolved processes. While the SHERPA-LO prediction stays below the data, the PYTHIA predictions agree within the error bars; the SHERPA-MC@NLO predictions describe the data well with the exception of the low-$E_T$ phase space in the direct process, where we see an undershoot of about 20%. As the cuts on the pseudo-rapidity select the forward region, this observable is probably sensitive to additional radiation from underlying events, the used PDFs, and to the photon splitting in the parton shower. Fig. 8 shows a similar situation for the $\eta$-dependence, where for the resolved processes both, PYTHIA and SHERPA-MC@NLO, agree with data but for direct processes we see an undershoot with SHERPA.

We finish this section with the distributions in $x_\gamma$ for low and high leading jet transverse energy in Fig. 9. Opposed to the modelling for LEP, there is no undershoot visible at the transition from direct to resolved processes. SHERPA undershoots the data in the region where both $x_\gamma$ and $E_T$ are small, however this can be attributed to the missing tuning of the MPIs as the same region is fairly well described by PYTHIA. HERWIG is again compatible with the leading order simulation from SHERPA.

## 5 Predictions for EIC

For the planned EIC we present predictions for electron-proton beams with 18 and 275 GeV beam energies, respectively, similar to the study in [62]. We cluster jets with the anti-$k_T$ algorithm with $R = 1.0$ and demand at least one jet with $E_T > 6$ GeV. Looking at inclusive (di-)jet observables in Fig. 10, we see a similar behaviour in the comparison between the generators, where SHERPA-LO yields the smallest cross-section and PYTHIA-LO the largest, while HERWIG delivers a significantly different shape of the $x_\gamma$ distribution. The $K$ factor in these observables is roughly 50%, again hinting at the real correction and the phase space driving the correction at NLO. The PYTHIA LO prediction deviates another 50% from the NLO-accurate prediction,

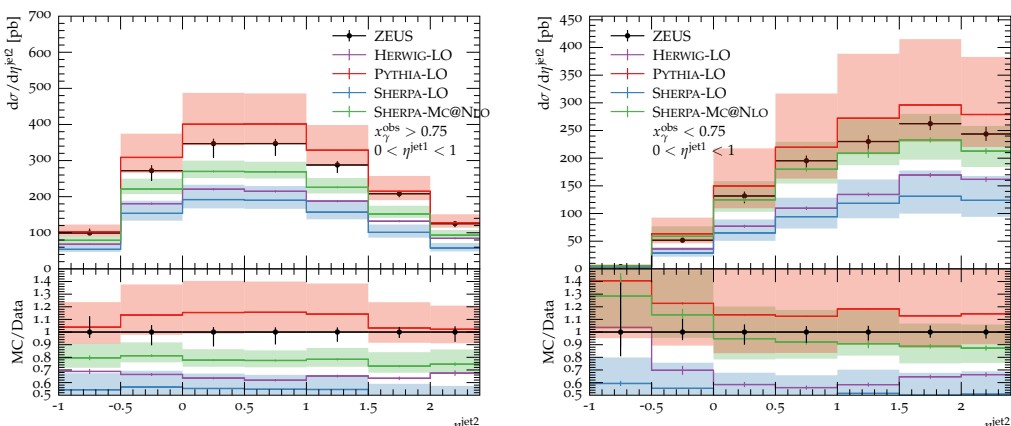

**Figure 8:** Distributions of sub-leading jet pseudo-rapidity $\eta^{\mathrm{jet2}}$ for $x_\gamma > 0.75$ (left) and $x_\gamma < 0.75$ (right) with leading jet pseudo-rapdity in $0 < \eta^{\mathrm{jet1}} < 1$ from ZEUS [61], compared to Leading Order simulations by HERWIG, PYTHIA and SHERPA and MC@NLO-accurate simulations by SHERPA.

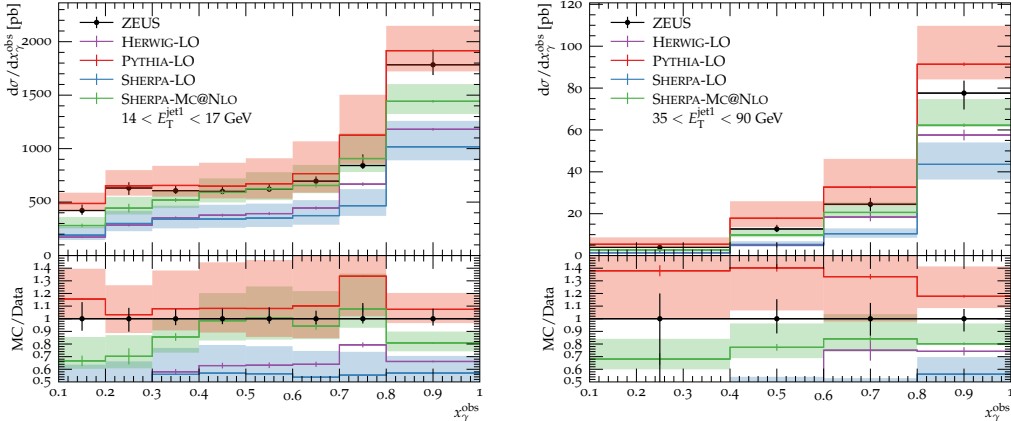

**Figure 9:** Distributions of $x_\gamma$ for leading jet transverse energies in $14 < E_T^{\mathrm{jet1}} < 17$ GeV (left) and $25 < E_T^{\mathrm{jet1}} < 90$ GeV (right) from ZEUS [61], compared to Leading Order simulations by HERWIG, PYTHIA and SHERPA and MC@NLO-accurate simulations by SHERPA.

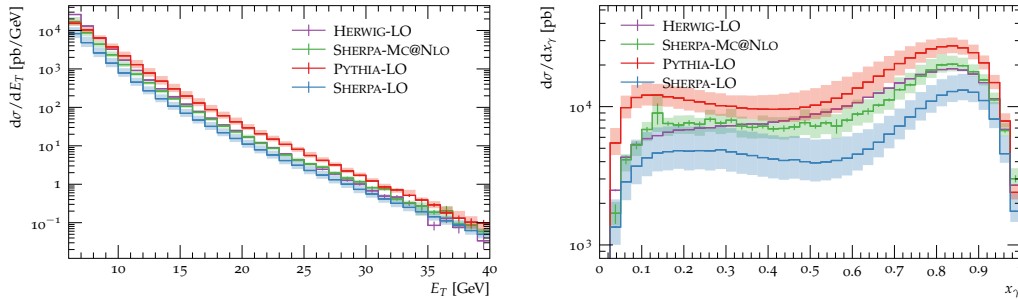

**Figure 10:** Predictions of transverse jet energy $E_T$ (left) and $x_\gamma$ (right) for jet production at the Eic, comparing to Leading Order simulations by Herwig, Pythia and Sherpa and Mc@Nlo-accurate simulations by Sherpa.

which can partially be explained by the different PDF sets, $\alpha_S$ value and the other differences as pointed out in Subsec. 2.4. This means that, going towards highest possible precision, the perturbative accuracy needs to be improved further and the non-perturbative effects need to be constrained by data. Publicly available Rivet analyses for the relevant data from Hera and Lep are crucial to reach the latter goal. While there have beens some recent progress on porting the existing analyses to Rivet framework, there still are some shortages related to MPI constraints and virtuality modelling.

To study the event shapes in more detail, we present predictions for transverse thrust $T_\perp$ and transverse sphericity $S_\perp$, defined as

$$T_\perp = \max_{\vec{n}_T} \frac{\sum_i |\vec{p}_{T,i} \cdot \vec{n}_T|}{\sum_i \vec{p}_{T,i}} \tag{5.1}$$

$$S_\perp = \frac{2\lambda_2}{\lambda_1 + \lambda_2} \tag{5.2}$$

in Fig. 11. Here $n_T$ is the transverse-thrust axis that maximize the quantity and $\lambda_{1,2}$ are the eigenvalues of the transverse linearised sphericity tensor $\mathbf{S}_{\alpha\beta}$ defined as

$$\mathbf{S}_{\alpha\beta} = \frac{1}{\sum_i |\vec{p}_{T,i}|} \sum_i \frac{1}{|\vec{p}_{T,i}|} \begin{pmatrix} p_{i,x}^2 & p_{i,x}p_{i,y} \\ p_{i,y}p_{i,x} & p_{i,y}^2 \end{pmatrix} \tag{5.3}$$

with $i$ summing over the momenta in the final state and $\alpha$, $\beta$ over $1, 2$. In both observables, Pythia predicts slightly more isotropic events than Sherpa, which might again be caused by a larger number of MPIs modelled within Pythia. Comparing the two Sherpa predictions, we again observe a sizeable $K$ factor and a shift towards more isotropic events. While using a higher value for $\alpha_S(M_Z)$ in Pythia as a proxy for the $K$ factor seems to work quite well and gives similar cross sections, the uncertainties from scale variations are considerably smaller at NLO.

As a last observable, in Fig. 12 we look at the charged particle multiplicity in the detector acceptance range $|\eta| < 4$ again for events that contain at least one jet with $E_T > 6$ GeV and see large disagreement between Pythia and Sherpa. Even though MPIs do not play a huge role when studying high-$p_T$ observables, such as jets, they do come into play when studying event structure in more detail. The result shows less hadrons being generated in Sherpa than in Pythia or Herwig and it means that a careful study of MPIs and hadronisation is necessary to correctly simulate these observables. As discussed before, while the perturbative accuracy is under good control, corrections due to non-perturbative effects rely on data being made available.

# 6 Conclusion

We presented a comparison of the three general purpose event generators, Herwig, Pythia, and Sherpa, and contrasted the results against experimental data from LEP and HERA colliders. While the starting point and theoretical ingredients are similar in each of the generator, the default inputs and differences in phenomenological modelling do result in significant differences for the considered observables. In particular

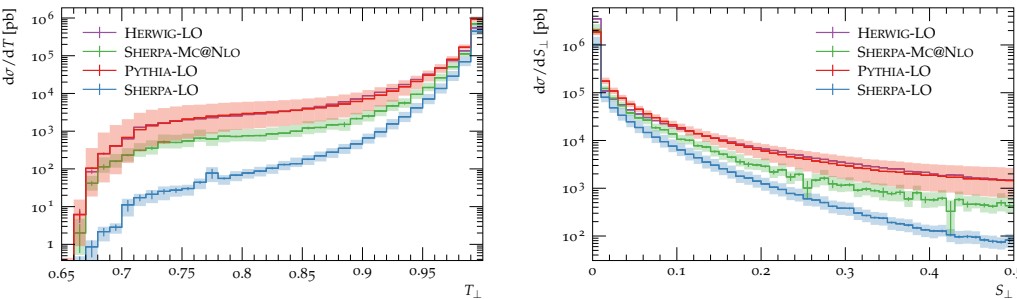

**Figure 11:** Predictions of transverse thrust $T_\perp$ (left) and transverse sphericity $S_\perp$ (right), for jet production at the EIC, comparing to Leading Order simulations by HERWIG, PYTHIA and SHERPA and MC@NLO-accurate simulations by SHERPA.

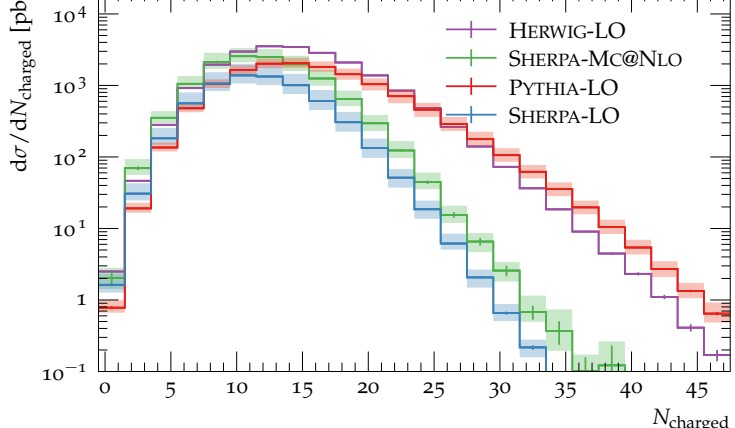

**Figure 12:** Predictions of charged particle multiplicity $N_{\text{charged}}$ for events with a jet with $E_{\text{T}} < 6$ GeV at the EIC, comparing to Leading Order simulations by HERWIG, PYTHIA and SHERPA and MC@NLO-accurate simulations by SHERPA.

we compared to data for dijet photoproduction analyzed by the OPAL and ZEUS collaborations, seeing an overall good agreement with the data for the SHERPA-MC@NLO and the PYTHIA simulations. At leading order, to which the HERWIG simulations are currently limited, and consistent with SHERPA at leading order, shapes are roughly consistent with the data, however normalizations lack a significant $K$-factor. Furthermore, we presented predictions for the upcoming EIC for inclusive QCD observables and event shapes for events containing at least one high-$p_\mathrm{T}$ jet. We found significant differences in observables sensitive to the underlying event modelling and concluded that by having more experimental data in RIVET framework would allow for further tuning of the parameters related to MPI generation such discrepancies could be largely resolved.

Being the dominant production mechanism for hadronic final states, more work is needed in preparation for precision phenomenology for the EIC to understand the different regimes and a coherent modelling of these, including development of the relevant infrastructure in all multi-purpose event generators. Open questions remain like the transition region between DIS and photoproduction at virtualities of $Q^2 \approx 1~\mathrm{GeV}^2$ or the direct and resolved regimes of the photon, and, more importantly, better fits to the parton content of the photon which are needed as a key ingredient for precision phenomenology at the EIC.

## Acknowledgements

We thank the organizers and the staff at the PhysTev 2023 conference, where this study was started. P.M. is supported by the STFC under grant agreement ST/P006744/1. I.H. has been supported by the Academy of Finland, project 331545, and funded as a part of the Center of Excellence in Quark Matter of the Academy of Finland, project 346326. The reported work is associated with the European Research Council project ERC-2018-ADG-835105 YoctoLHC.

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
