# Peer review of "Photoproduction in general-purpose event generators"

_SciPost Physics Community Reports_

## Round 2 · Referee Report · Anonymous (Referee 1) · 2024-8-29

Report

With this article the authors present a set of predictions for jet final states produced in photo-production obtained with the three multi-purpose event generators Herwig, Pythia and Sherpa. These predictions get compared against a selection of measurement results from LEP and HERA experiments. Furthermore, predictions corresponding to photo-production processes at the future EIC machine are presented.

Photo-production is certainly a very relevant production channel that received comparably little attention over the past years, but experiences revived interest recently, motivated by the upcoming EIC experiments. Thus a comparison of standard tools used for photo-production simulations, thereby providing a benchmark for future developments, is certainly interesting and timely. However, in its current form I cannot recommend the article for publication in SciPost Physics Community Reports, given the paper lacks sufficient detail and content. Furthermore, parts of the article are written in an almost careless, unambitious and rushed manner, requiring significant improvement before considering publication.

With respect to the lack of sufficient content, the actual and obviously relevant differences between the generator models are neither clearly worked out and described in Sec. 2. nor are later on in Sec. 3 observed significant differences between the various predictions carefully explained and traced back to the different implementation and parameter choices in the generators. For example, the authors repeatedly claim that a dominant part of the differences originate from the MPI activity. However, no evidence is provided, e.g. via results corresponding to different tunes in Pythia or at least rough variations of relevant parameters in the other generators. Similar arguments apply for the differences in the employed PDFs, the values for the running coupling etc. More systematic comparisons should be straight forward for the generator authors and in fact would provide very useful insight and information for the reader. Further, these studies could potentially provide guidance for complementary measurements and analyses helping to further constrain the models. Further, the authors do not convincingly describe the origin of the vastly different systematic uncertainties quoted for the four generator predictions, leaving aside that these are fully missing for Herwig throughout.

Furthermore, the initial comparison at the parton level presented in Figs 1 & 2 fully lacks Herwig predictions and also lacks an attempt to more quantitatively understand the deviations observed upon inclusion of the parton showers in Pythia and Sherpa.

In terms of editorial deficiencies, in particular the introduction is poorly edited language wise, with lots of missing articles, and hard to understand sentences.
Consider for example just the first two sentences of the second to last paragraph:

"The development of the current event generators have mainly been driven by the studies of proton-proton (pp) collisions at the LHC. The provided luminosities have enabled measurements to reach accuracies of a few
percent-level, necessitating persisting theoretical computations of the differential cross-sections to matching or higher precision"

The introduction to Sec. 2 lacks basically any reference to relevant prior works, are clear omission. There are plenty of singular/plural mistakes. Eq. (5.3) contains a typo. Many figures are rather unsatisfying, featuring legends overlapping with curves, lines in ratio plots falling outside the plot range.

Overall I cannot recommend the article in its present form for publication.

Recommendation

Reject

---

## Round 2 · Referee Report · Anonymous (Referee 2) · 2024-10-4

Report

Apologies for the delay in providing this report.

As efforts especially towards EIC ramp up, I find a study of this kind timely. I also appreciate the cross-collaboration effort to present and contrast different treatments, giving an overview and instantaneous snapshot of the current state of the art, accompanied by remarks on what can be expected in the near future. I think such an overview has been missing and will be valuable to the photo-production community.

A topical overview like this is a good opportunity for authors to explain and illustrate the effects of modelling inputs, constraints, capabilities, limitations, and, consequently, uncertainties, within a limited physical context, directed at a specific community. This is often less confusing to that community, and of higher direct use, than the general documentation and physics papers of the models in question. When authors following several different approaches gang up, as here, a truly useful review can result.

Studying the first referee report, I was afraid that the authors had missed this opportunity and fallen into the trap of providing a “user-level” comparison, without insights into how and why differences arise. But I must honestly say I do not find this to be the case. (Perhaps the authors have updated their manuscript since the first report?) Starting from the perfect agreement observed at the “outset” in Fig 1 (which is nice to see), the authors consider successively further components of the modelling, in successively more complicated physics contexts, and in each case discuss the differences. E.g., just when I would have liked an overview of the differences between the default settings of the generators for photoproduction, one was provided in the form of Table 1. And when, in that table, I did not immediately know the difference between LL and NLL flux, this was explained in eq.(2.4), along with physical insight spelling out what to expect in terms of cross sections. The (significant) differences between alphaS parametrisations are also commented upon. And finally, not being an expert in this domain, I did not know what all the different PDF choices really amounted to. And there was the comprehensive fig. 3. This is exactly what I need to see. So far so good.

In Figs 4-7, I do agree that the differences are not well explained. There is just the comment that these differences “build up from varying inputs”. It would be nice to understand if there is any specific driver, e.g., of the Pythia predictions being so large. Is it alphaS? Is it radiation kernels? Is it MPI? PDFs? All of these things can be varied within the model (or even switched on/off?) to see how significant each one is? That would also give context to put the LO -> NLO change in Sherpa on firmer ground.

Same comment for figs. 8-9. There is just the statement that “this observable is probably sensitive to additional radiation from underlying events, the used PDFs, and to the photon splitting in the parton shower.” Why not vary those things or switch them on/off to see explicitly?

I wager such additional checks would also help bring a firmer understanding to the EIC predictions presented in the last section, so this is my main request of the authors.

Requested changes

The requested changes are contained in the report. I repeat them here for clarity:

  1. In Figs 4-7, I do agree that the differences are not well explained. There is just the comment that these differences “build up from varying inputs”. It would be nice to understand if there is any specific driver, e.g., of the Pythia predictions being so large. Is it alphaS? Is it radiation kernels? Is it MPI? PDFs? All of these things can be varied within the model (or even switched on/off?) to see how significant each one is? That would also give context to put the LO -> NLO change in Sherpa on firmer ground.

  2. Same comment for figs. 8-9. There is just the statement that “this observable is probably sensitive to additional radiation from underlying events, the used PDFs, and to the photon splitting in the parton shower.” Why not vary those things or switch them on/off to see explicitly?

  3. Use the lessons from points 1 and 2 to bring a firmer understanding to the EIC predictions presented in the last section.

Recommendation

Ask for major revision

---

## Editorial Decision

awaiting_resubmission